# Fluorescence Microscopy: Determination of Meropenem Activity against *Klebsiella pneumoniae*

**DOI:** 10.3390/antibiotics12071170

**Published:** 2023-07-10

**Authors:** Kamilla N. Alieva, Maria V. Golikova, Anastasia A. Kuznetsova, Stephen H. Zinner

**Affiliations:** 1Department of Pharmacokinetics & Pharmacodynamics, Gause Institute of New Antibiotics, 11 Bolshaya Pirogovskaya Street, 119021 Moscow, Russia; golikovaka@gmail.com (M.V.G.); biolisichka@gmail.com (A.A.K.); 2Department of Medicine, Harvard Medical School, Mount Auburn Hospital, 330 Mount Auburn Street, Cambridge, MA 02138, USA; szinner@mah.harvard.edu

**Keywords:** meropenem, *K. pneumoniae*, antibiotic resistance, fluorescence microscopy, acridine orange, OXA-48-carbapenemases, antibiotic pharmacodynamics, susceptibility testing

## Abstract

The development and implementation of diagnostic methods that allow rapid assessment of antibiotic activity against pathogenic microorganisms is an important step towards antibiotic therapy optimization and increase in the likelihood of successful treatment outcome. To determine whether fluorescence microscopy with acridine orange can be used for rapid assessment (≤8 h) of the meropenem activity against *Klebsiella pneumoniae*, six isolates including three OXA-48-carbapenemase-producers were exposed to meropenem at different levels of its concentration (0.5 × MIC, 1 × MIC, 8 or 16 µg/mL) and the changes in the viable counts within 24 h were evaluated using fluorescence microscopy and a control culture method. The approach was to capture the regrowth of bacteria as early as possible. Within the first 8 h fluorescence microscopy allowed to categorize 5 out of 6 *K. pneumoniae* strains by their meropenem susceptibility (based on the MIC breakpoint of 8 mg/L), but meropenem activity against three isolates, two of which were OXA-48-producers, could not be accurately determined at 8 h. The method proposed in our study requires improvement in terms of accelerating the bacterial growth and regrowth for early meropenem MIC determination. Volume-dependent elevation in meropenem MICs against OXA-48-producers was found and this phenomenon should be studied further.

## 1. Introduction

Optimization of antibiotic therapy is an important problem of modern health care. To select individual regimens for the treatment of bacterial infections information about the sensitivity of the pathogen to a particular antibiotic or antibiotic combination is required. In case of severe infectious diseases, not only the accuracy but the speed of antimicrobial susceptibility testing (AST) becomes critical, since the prompt administration of effective antimicrobial therapy can save the patient’s life. Cultural methods for assessing the minimal inhibitory concentration (MIC) of an antibiotic against a pathogenic microorganism such as microdilution, disk diffusion, or E-test require long-term cultivation of microorganisms, and their results can be obtained no earlier than a day after the start of the evaluation [1].

The development and implementation of diagnostic methods that allow rapid assessment of antibiotic activity against pathogenic microorganisms (optimally—within a few hours after the start of the analysis) is an important step towards antibiotic therapy optimization and increase in the likelihood of successful treatment outcome. Various rapid AST methods have been proposed, including genomic and phenotypic approaches [2,3]. DNA-based tests could provide only limited information on antibiotic susceptibility, since the knowledge of the resistance mutations and genes is limited and some mechanisms, which allow bacteria to withstand exposure to antibiotics, are non-genetic [4]. Therefore, research efforts are currently focused on phenotypic rapid AST assays. A notable part of the studies in this area are devoted to the development of optical methods using fluorescence-based assays. Flow cytometry with fluorescent viability markers, in which the changes in bacterial morphology and physiology caused by antibiotics are detected, is actively developed as it is considered a promising approach for rapid AST [5,6,7,8,9,10,11,12,13]. In the studies using flow cytometry, disruption of cell vitality is analyzed based on monitoring of changes in size, shape, cytoplasmic volume of bacteria [7,10,11,12], bacterial membrane alteration [6,7,8,11,13], and other changes in cell physiology, such as carbonylation [11]. This approach allowed to obtain AST results after 1–3 h, and these results were generally in good agreement with those obtained by culture methods.

However, for a number of insidious bacteria, the reliability of AST estimates based on early cell damage or death may be questionable. Carbapenemase-producing *K. pneumoniae* (CPKP) is among such pathogens. Mulroney et al. studied meropenem activity against *K. pneumoniae* with 30 min exposure followed by flow cytometry analysis [12]. With 48 isolates, a strong positive correlation was observed between the results obtained by cytometry-assisted susceptibility test and broth microdilution. However, majority of the studied isolates were either highly susceptible carbapenemase non-producers (MIC ≤ 0.25 mg/L) or resistant carbapenemase producers (MIC ≥ 8 mg/L) and the established relationship was based primarily on the results obtained with them. If the isolates with MICs of meropenem between these values (0.25 ≤ MIC ≤ 8 mg/L) are considered separately, proposed method for rapid AST might not be reliable enough: MIC values of meropenem against carbapenemase-producers obtained by this method differed from the broth microdilution results by 2–8 times, and two non-susceptible strains were defined as susceptible due to this difference in MIC. This suggests that special attention should be paid to the sensitivity assessment of such isolates. In their case, capturing the regrowth, rather than cell viability impairment or death, may be a better approach for rapid AST.

In addition to revisiting the time needed for a reliable MIC assessment for some bacteria, more consideration should probably be given to rapid AST methods, which would be more accessible than flow cytometry in terms of required equipment and reagents. Fluorescence microscopy [14] with a readily available fluorophore acridine orange [15] may be considered among such methods. To date it has been used mainly for microscopic examination of clinical specimens and detection of viable bacteria using manual [16,17,18,19,20,21] or semi-automated methods [22]. Acridine orange has also been studied as an indicator of bacterial susceptibility to antibiotics. It was used to assess fleroxacin activity against *Pseudomonas fluorescens* biofilms with scanning confocal laser microscopy [23]. A flow cytometry method with acridine orange for rapid evaluation of *Escherichia coli* susceptibility to gentamicin was also described [5].

To determine whether fluorescence microscopy with acridine orange can be used for rapid assessment of the meropenem activity against *Klebsiella pneumoniae*, six isolates, including three OXA-48-carbapenemase-producers, were exposed to meropenem at different levels of its concentration and the changes in the viable counts within 24 h were evaluated using fluorescence microscopy and a control culture method.

## 2. Results

At the first stage of the study, we aimed to verify the applicability of fluorescence microscopy for assessment of viable bacterial counts in time-kill experiments. These experiments were carried out with six clinical and two reference *K. pneumoniae* strains, with reference ones used as a control. Viable cells of all strains had a well-defined fluorescence, which varied from green to red. Cells in the samples not exposed to meropenem had a characteristic shape and arrangement inherent in *K. pneumoniae* throughout the observation period: small rounded rods were arranged singly, in pairs and less often in short chains. As bacteria were not fixed at the slide, they floated in the liquid, moving with its slight fluctuations. The mucoid isolate *K. pneumoniae* 1128 was an exception: at 8 and 24 h its cells adhered to the surface of the slide and were completely immobile. Among the viable cells exposed to the antibiotic, there was a slight heterogeneity in the forms and intensity of fluorescence.

The changes in the numbers and characteristics of bacteria observed by microscopy in the absence and presence of meropenem can be seen in the example of *K. pneumoniae* 1676 (Figure 1). In the absence of the antibiotic, the number of cells steadily increased; they had a typical shape and a uniform, predominantly green fluorescence. Cells exposed to 0.5 × MIC of meropenem remained viable after 4 h, and the proportion of cells with red fluorescence became bigger than the proportion of cells with green fluorescence. Subsequent regrowth led to a significant increase in the bacterial numbers and the proportion of cells with green fluorescence. In addition to typical variants, cells of a more rounded shape and with brighter fluorescence were present in the fields. At the higher antibiotic concentrations, starting from 4 h single cells could come across in the field of view or their complete absence was observed.

Fluorescence microscopy revealed cell size and shape alteration in both carbapenemase non-producers and carbapenemase-producers in the presence of meropenem (Figure 2 and Figure 3). As could be seen from the figures, at 4–24 h after the start of meropenem exposure, enlarged cells with distorted form appeared along with cells of typical size and form. The most prominent cell size alterations were observed with *K. pneumoniae* ATCC BAA-1904 at 4 h (Figure 3). Restoration of the visual appearance of the bacterial population at 24 h, relative to that at the beginning of the experiment, was observed with carbapenemase-producing strains, while with carbapenemase non-producing strains cell deformation usually remained till the end of the observation period.

Time courses of carbapenemase non-producing *K. pneumoniae* 844, *K. pneumoniae* 1676, and *K. pneumoniae* 2286 in the absence of the antibiotic and exposed to meropenem are shown in Figure 4. The viable cell numbers of each strain, determined by agar plating and microscopy (green or total fluorescence), practically coincided throughout the entire observation period, while the number of cells determined based on red fluorescence only could be close to the results of these measurements, or differ significantly. Red fluorescence usually accompanied the secondary growth of bacteria, but was often absent after the accumulation of high cell density.

In the absence of meropenem, bacterial counts exceeded 8 log (CFU/mL) by 8 h, while at the sub-MIC concertation of meropenem, moderate growth (up to 6.9–7.6 log (CFU/mL) depending on the evaluation method) and regrowth (up to 5.5 log (CFU/mL)) was noted with *K. pneumoniae* 1676 and *K. pneumoniae* 2286, respectively. With *K. pneumoniae* 844 exposed to 0.5 × MIC of meropenem, no regrowth was noted by 8 h, but at 24 h its bacterial density was similar to the one not exposed to the antibiotic. At meropenem concentration of 1 × MIC or higher, viable counts of all three strains decreased below the limit of detection and remained at this level until the end of the observation.

Results obtained with OXA-48-producing *K. pneumoniae* 1128, *K. pneumoniae* 1170, and *K. pneumoniae* 1456 under the same conditions were similar in terms of the convergence of data obtained by microscopy and culture methods (Figure 5). However, the MIC values of meropenem against these strains, obtained by the macrodilution method, were higher than the corresponding values obtained by the microdilution method: 1 versus 0.5 mg/L for *K. pneumoniae* 1128, >8 versus 4 mg/L for *K. pneumoniae* 1170, and >8 versus 2 mg/L for *K. pneumoniae* 1456. Moreover, according to the results of macrodilution, meropenem MICs against *K. pneumoniae* 1170 and *K. pneumoniae* 1456 exceeded the clinical breakpoint of 8 mg/L, which would render these strains resistant to the antibiotic. It should also be noted that in the presence of meropenem at concentrations above 0.5 × MIC, at 8 h regrowth could be observed only with *K. pneumoniae* 1456, while the other two isolates regrew later.

Intensive regrowth of OXA-48-producing *K. pneumoniae* isolates was accompanied by a 67–95% decrease in the initial meropenem concentrations by 24 h. After the same time, the loss of meropenem in the experiments with carbapenemase non-producers was 27–54%. This difference suggests that the regrowth of the OXA-48-producers was accompanied by an intensification of the production of carbapenemases, which added additional destruction of antibiotic molecules in addition to thermal decomposition. Of note, after 8 h of incubation, meropenem concentrations decreased by a similar amount for OXA-48-producers and carbapenemase non-producers (14–27% and 10–32%, respectively), which may indicate that the main effect of carbapenemases occurred after 8 h.

Time courses of reference stains, *K. pneumoniae* ATCC 700603 (carbapenemase non-producing) and *K. pneumoniae* ATCC BAA-1904 (KPC-producing), in the absence of the antibiotic and exposed to meropenem are shown in Figure 6. Results with these strains, similarly to those with clinical isolates, indicate good convergence in the assessment of the number of viable cells using the microscopy method and the culture method throughout the entire observation period.

High convergence between the results, obtained by microscopy (green and total fluorescence) and the culture method, was observed regardless of time, strain, and antibiotic concentration (Figure 7). Combined data could be fitted by linear function (Equation (1)) with the high squared correlation coefficients (*r*^2^ = 0.95 for both). Data variability with red fluorescence resulted in weaker correlation (*r*^2^ = 0.61). However, cells with red fluorescence can be in the majority at 4 and 8 h, and should not be excluded. Therefore, the assessment of the viable counts of *K. pneumoniae* exposed to meropenem by total fluorescence seems to be the most correct among the three options.

The possibility of early meropenem MIC determination (within 8 h) was evaluated by the regression analysis of time-kill data, obtained with the clinically relevant OXA-48 producing and carbapenemase non-producing strains. If the method proposed in this study works as expected, bacterial counts at 8 and 24 h should be equally high in the absence of the antibiotic and equally low at the antibiotic concentration of 1 × MIC and above. In the presence of the antibiotic at concentrations below the MIC, a moderate number of bacteria at 8 h and a high number of bacteria at 24 h should be expected. Such results would likely be described by the sigmoid function. This assumption turned out to be partially true for the data obtained in the present study. Regrowth in carbapenemase non-producing *K. pneumoniae* 844, *K. pneumoniae* 1676, and *K. pneumoniae* 2286 was observed only at antibiotic concentrations below 1 × MIC, and usually already appeared at 8 h. Thus, the combined data for 8 and 24 h at various C/MICs could be fitted with the sigmoid function (Equation (2)) with *r*^2^ = 0.91 and 0.89 for agar plating and total fluorescence, respectively (Figure 8). Corresponding data for OXA-48 producers could not be fitted with this function: Points were scattered horizontally due to a change in the meropenem activity in an increased volume of the medium and vertically due to a significant gap between the viable counts at 8 and 24 h.

## 3. Discussion

In this work we proposed a method with readily available reagents and simple sample preparation, which can be used for assessment of the number of viable bacteria in the presence of an antibiotic in real time. According to the assessment of the *K. pneumoniae* viable counts both in the absence and presence of meropenem, fluorescence microscopy with acridine orange provides results similar to the results of agar plating, when green and total fluorescence are taken into the account (*r*^2^ = 0.95). Based on the regression analysis data, red fluorescence estimation was not suitable for such an assessment in the case of the method used and the studied antibiotic and microorganism. Given that red fluorescence is usually exhibited by actively dividing cells [24], accounting it specifically may be important for evaluating the activity of bacteriostatic antibiotics. Additional research is necessary to clarify this.

Within the first 8 h fluorescence microscopy with acridine orange allowed to categorize 5 out of 6 *K. pneumoniae* strains by their meropenem susceptibility (based on the MIC breakpoint of 8 mg/L). Only one of the isolates, *K. pneumoniae* 1170, regrew after 8 h when exposed to 8 mg/L of meropenem. However, if regrowth patterns at all antibiotic concentrations (0.5×, 1 × MIC and 8 or 16 mg/L) are taken into account, meropenem activity evaluation at 8 and 24 h matched only for two carbapenemase non-producing isolates and one CPKP isolate. Therefore, the 8-h susceptibility estimate under the current conditions may be inaccurate, especially if the tested strain is CPKP. Importantly, bacterial regrowth after 8 h could start from very low levels below the limit of detection (3.3 log (CFU/mL)). These small quantities of surviving bacteria could be overlooked by methods focusing on antibiotic-induced bacterial killing prediction based on early disturbances in cell metabolism [6,7,8,10,11,12,13]. Therefore, such methods may be not optimal for bacteria with unpredictable killing-regrowth patterns, such as CPKP.

We observed the cell size and shape alteration, which occurred both in carbapenemase non-producing and carbapenemase-producing *K. pneumoniae* strains in the presence of meropenem (Figure 2 and Figure 3). These changes may be associated with modifications in the physiological state of the bacterial cells under the antibiotic exposure [25]. For example, an increase in the cell volume can decrease the antibiotic concentration inside a cell, thereby promoting cell growth. In addition, an increased cell volume along with relative surface area of the cell can reduce the flux of antibiotics into the cell.

The AST method proposed in our study requires improvement in terms of accelerating the results. We consider the use of a more nutritious culture media as a possible way to hasten the bacterial growth, if the use of these media does not distort the results of the MIC assessment. The approach of using rich nutrient media (Todd-Hewitt broth) to speed up the AST has been successfully used before [26]. Such an approach may work with carbapenemase-producing *K. pneumoniae* strains: Although faster growth would likely stimulate the effect of the antibiotic, the production of carbapenemase can also increase, which could balance the effect of the antibiotic and allow the obtaining of an early result.

Volume-dependent elevation in meropenem MICs against OXA-48-producing *K. pneumoniae* (2-fold for *K. pneumoniae* 1128, ≥2-fold for *K. pneumoniae* 1170 and ≥4-fold for *K. pneumoniae* 1456) is of interest and concern. According to macrodilution results, two of these isolates are resistant to meropenem, which is inconsistent with the microdilution results. Considering that the volume of fluid in the infection site may be significantly larger than the volume of the well of the microtiter plate, it is logical to assume that the macrodilution results may better reflect the potential antibiotic activity in humans. To our knowledge, studies aimed at evaluating the effect of volume on the MIC of antibiotics are few, and neither beta-lactams nor gram-negative bacteria have been studied in this aspect. Decrease in antibiotic activity with increase in the broth volume was observed before with vancomycin and teicoplanin and *Staphylococcus aureus*, and it was noted that MICs could be underestimated when evaluated by broth microdilution compared to macrodilution [27]. While the development of novel AST methods is usually aimed at using small volume samples, our results suggest that development in the opposite direction should also be taken into account.

Increase in MIC of beta-lactam antibiotics against beta-lactamase-producing bacteria can also occur at increased inoculum density (~5 × 10^7^ CFU/mL) and this phenomenon is also known as the “inoculum effect” [28,29,30,31,32]. The cause of it is not yet determined. It is assumed that this effect occurs due to a large population of bacteria quickly filling the limited volume with beta-lactamases, presumably after antibiotic-induced cell lysis in case of gram-negative bacteria, which decompose the target drug [28,30,31]. In the current study, we observed an increase in MIC accompanied by meropenem destruction, caused by an increase in the broth volume, but not the size of the *K. pneumoniae* inoculum. Such results indicate that the accumulation of beta-lactamases can be associated not only with the accumulation of large bacterial numbers, but with the activation of the mechanisms of cell interaction between each other or with the external environment, which induces the production of beta-lactamases. Considering both increase in volume or inoculum may lead to an increase in the beta-lactamase activity, we hypothesized that the MIC at increased volume may be close to the MIC at high inoculum (MIC_HI_). Since we had the results of meropenem MIC_HI_ evaluation for all the strains (8, 4, and 2 mg/L for *K. pneumoniae* 844, 1676, and 2286, respectively, and 16, 32, and 64 mg/L for *K. pneumoniae* 1128, 1170, and 1456, respectively), the relationship between the viable counts at the end of observation and C/MIC_HI_s could be evaluated and compared to the relationship with C/MICs (Figure 9). For carbapenemase-producers, C/MIC_HI_ values, which were 8-fold (*K. pneumoniae* 1128) and 32-fold (*K. pneumoniae* 1170 and *K. pneumoniae* 1456) lower than C/MICs, shifted to the left along the x-axis, and the absence of bacterial growth corresponded to C/MIC_HI_ ≥ 1. With C/MIC_HI_, combined data for carbapenemase producers and non-producers could be fitted with the sigmoid function (Equation (2)) with *r*^2^ = 0.86, while with C/MIC, the correlation was very weak due to the large scatter of points (*r*^2^ = 0.23). These findings support the assumption that the volume-dependent MIC elevation (“volume effect”) may have similar underlying mechanisms with the “inoculum effect”. More research needs to be done to understand which antibiotics and bacteria may exhibit this phenomenon, what is the cause of it, and whether it is indeed somehow related to the “inoculum effect”.

The study has several limitations: it did not include many bacterial strains and species. The use of a greater number of bacterial strains and other beta-lactam antibiotics could generalize the current conclusions. In addition, it could be of interest to investigate the possibility of early prediction of the antibiotic activity against clinical *K. pneumoniae* strains with low meropenem MICs. Last but not least, the existence of bacterial strains that are susceptible to antibiotics but can carry the silent (cryptic) resistance genes was numerously reported [33,34]. Such examples are of particular importance, as the presence of the silent resistance genes in infectious agents can be a risk factor of unexpected treatment failures. It can be assumed that the “volume effect” observed in the current study may be a consequence of the presence of such genes in bacteria. Further evaluation of that effect may provide an opportunity to reveal silent resistance patterns of strains initially susceptible to antibiotics. The cryptic genetic mechanisms are a complex phenomenon that requires thorough investigation.

## 4. Materials and Methods

### 4.1. Antimicrobial Agent, Bacterial Isolates, and Susceptibility Testing

Meropenem powder was purchased from Sigma-Aldrich (St. Louis, MO, USA). Six clinical isolates of *K. pneumoniae* collected from the clinical samples of ICU patients admitted to the Moscow and Saint-Petersburg hospitals in the year 2021 were used in the study. Three of the six strains were carbapenemase-negative by PCR: 844 (meropenem MIC = 8 mg/L), 1676 (meropenem MIC = 4 mg/L) and 2286 (meropenem MIC = 2 mg/L). The other three strains were OXA-48 carbapenemase-positive by PCR: 1128 (meropenem MIC = 0.5 mg/L), 1456 (meropenem MIC = 2 mg/L), and 1170 (meropenem MIC = 4 mg/L). In addition, two reference *K. pneumoniae* strains were used in the study: carbapenemase-negative ATCC 700603 (meropenem MIC = 0.03 mg/L) and KPC carbapenemase-positive ATCC BAA-1904 (meropenem MIC = 8 mg/L). The reference strains were used as controls in susceptibility testing and time-kill experiments. MICs were determined by the broth microdilution technique at an inoculum size of 5 × 10^5^ CFU/mL [1].

The verification of fluorescence microscopy applicability for the assessment of viable bacterial counts in the time-kill experiments was conducted both with clinical and reference (used as controls) *K. pneumoniae* strains.

### 4.2. Operational Procedure Used for Early Determination of Meropenem Activity

The broth macrodilution method was used for early determination of meropenem activity against *K. pneumoniae*. Flasks with 220 mL of Muller-Hinton broth (Becton Dickinson, Holdrege, NE, USA) without antibiotic and containing 0, 5 × MIC, 1 × MIC, and 8 mg/L (clinical MIC breakpoint) of meropenem were inoculated with 5 × 10^5^ CFU/mL of an 18h bacterial culture and incubated at 37 °C. Each flask with antibiotic and bacteria was continuously mixed by a magnetic stirrer to ensure uniform antibiotic distribution throughout the broth volume. Since the MIC value of meropenem against *K. pneumoniae* 844 and *K. pneumoniae* ATCC BAA-1904 is 8 mg/L, 16 mg/L (2 × MIC) was used as the highest antibiotic concentration for these strains. The concentration (C) range of meropenem and the corresponding concentration-to-MIC ratios (C/MICs) for each strain are shown in Table 1. Each flask was sampled at 0, 4, 8, and 24 h of incubation.

### 4.3. Quantitation of the Antimicrobial Effect Using Agar Plating

For bacterial enumeration with the control agar plating method, samples of 100 µL were serially diluted as appropriate and 100 μL were plated onto tryptic soy agar (Becton Dickinson, Holdrege, NE, USA) plates, which were incubated at 37 °C for 24 h and screened visually for growth. The lower limit of accurate detection was 3.3 log (CFU/mL) (equivalent to 20 colonies per plate).

### 4.4. Quantitation of the Antimicrobial Effect Using Fluorescence Microscopy

Sample preparation for microscopy was chosen depending on the expected bacterial density. If the suspension had visible turbidity, a sample of 5 µL (a high turbidity suspension; expected log (CFU/mL) > 8) or 25 µL (expected log (CFU/mL) > 7) was placed on a microscope slide, air dried aseptically and resuspended in 5 µL of 0.01% acridine orange stain (solution in acetate buffer; pH = 4), then a cover slip was placed on top. The sample after resuspension was considered to be evenly distributed under the entire area of the cover slip in one layer.

If the suspension was transparent or had a slight opalescence, bacterial cells were concentrated by sedimentation: The 50 mL of suspension was centrifuged at 3600× *g* and 4 °C for 20 min, then the supernatant was removed and samples were taken from the sediment (~0.5 mL) for microscopy (25 µL) and inoculation on agar to assess cell concentration (100 µL, then serially diluted in saline as appropriate). Samples for microscopy were processed as described above. Sedimentation allowed to concentrate cells by an average of 70 times. The lower limit for determining the number of viable cells by microscopy after sedimentation was 3.6 log (CFU/mL).

Immersion fluorescence microscopy was performed using a Leica DM IL Led microscope (Leica Microsystems CMS GmbH, Wetzlar, Germany) equipped with the I3 filter system for “blue excitation” (BP 450–490 nm; mirror 510; LP515(S)) at 1000 × magnification. Viable cells stained with acridine orange had green, yellow, orange, or red fluorescence on a green or dark red background. Bacterial counts were estimated from a count of at least five randomly chosen microscope fields.

### 4.5. Determination of Meropenem Concentrations by High-Performance Liquid Chromatography (HPLC)

To determine how much the antibiotic degrades in the presence of carbapenemase non-producers and producers, flasks were sampled at 0, 4, 8, and 24 h and meropenem concentration was evaluated using HPLC. Isocratic separation was performed at 30 °C on a column Luna C18 (2) (250 mm × 4.6 mm, particle size 5 microns; Phenomenex, Torrance, CA, USA). The sample volume was 10 μL. The mobile phase (MP) consisted of a 50 μM solution of potassium dihydrogen phosphate, adjusted with phosphoric acid to pH 2.4 (buffer) and acetonitrile (volume ratio 83.5:16.5) at a MP rate of 1.0 mL/min. Detection was carried out using a UV detector at 304 nm (Waters 2489, Waters Associates, Milford, MA, USA). The calibration graphs were linear (*r*^2^ ≥ 0.99) in the range of meropenem concentrations from 1.0 to 1000.0 μg/mL. The relative standard deviations (*n* = 5) for the concentrations of meropenem 1000.0, 50.0, 5.0, and 1.0 μg/mL were 1.0, 1.2, 2.7, and 5.5%, respectively. The lower limit of the quantitative determination of meropenem was 1.0 μg/mL. The limit of detection was 0.3 μg/mL. A 150 μL sample of broth with antibiotic was placed in a 1.5 mL Eppendorf plastic tube, 150 μL of acetonitrile were added, shaken for 1 min, and centrifuged for 5 min at 13,000 rpm at 5 °C. A 150 μL sample of supernatant was placed in 1.5 mL Eppendorf plastic tubes, 300 μL of buffer was added, shaken for 1 min, and centrifuged for 5 min at 13,000 rpm at 5 °C. The supernatant was analyzed by HPLC.

### 4.6. Statistical Analysis

In time-kill experiments, bacterial count data were calculated as arithmetic mean ± standard deviations for three replicate experiments. Based on these data, kinetic growth and time-kill curves were constructed. Assuming that the coefficients of variation for log CFU/mL data were less than 10%, to facilitate the figure viewing, we did not place data point error bars in order to not interfere with the kinetic curves.

The viable counts obtained by microscopy versus viable counts obtained by agar plating plots were fitted by a linear equation:(1)Y=a+bx
where *Y* is viable counts obtained by microscopy, *x* is viable counts obtained by agar plating, and *a* and *b* are parameters.

The viable counts versus ratio of antibiotic concentration to MIC (C/MIC) or to MIC at an inoculum size of 10^7^ (MIC_HI_, C/MIC_HI_) curves were fitted by the sigmoid function:(2)$$Y=Y_{0}+a/\left\{1+exp\left[-\left(x-x_{0}\right)/b\right]\right\}$$
where *Y* is viable counts; *x* is C/MIC or C/MIC_HI_; *Y*_0_ and *a* are the minimal and maximal concentrations of viable cells, respectively; *x_0_* is *x* corresponding to *a*/2; and *b* is a parameter reflecting sigmoidicity.

All calculations were performed using SigmaPlot 12 software (Systat Software Inc., headquartered in San Jose, CA, USA).

## 5. Conclusions

The findings of the current study suggest that fluorescence microscopy with acridine orange allows early determination of carbapenemase non-producing *K. pneumoniae* susceptibility to meropenem within 8 h (based on the meropenem MIC resistance breakpoint of 8 mg/L); MICs determined using the macrodilution technique were in concordance with those determined using the traditional microdilution method. Two out of three OXA-48 producing strains were characterized with significantly higher meropenem MICs determined by the macrodilution method, compared to corresponding values determined by the microdilution technique. The “volume effect” on the meropenem MIC may exist and it seems to be of a similar nature as an “inoculum effect”. This assumption was supported by a strong “C/MIC_HI_—bacterial counts at 24 h” relationship that was described with a sigmoid function. Considering that the volume of fluid in the infection site may be significantly larger than the volume of the well of the microtiter plate, it could be hypothesized that the macrodilution results may better reflect the potential antibiotic activity of meropenem against carbapenemase producers in humans. In addition, meropenem activity against CPKP could not be accurately determined at 8 h. The AST method proposed in our study requires improvement in terms of accelerating the bacterial growth and regrowth for early (≤8 h) meropenem MIC determination.

## Figures and Tables

**Figure 1 antibiotics-12-01170-f001:**
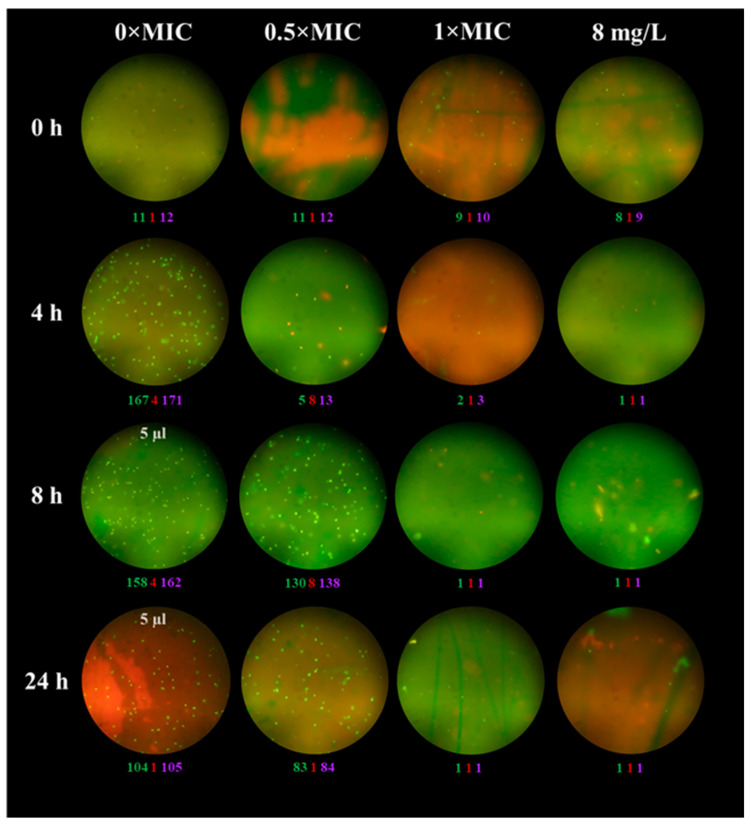
Typical microscope fields of *K. pneumoniae* 1676 exposed to 0×, 0.5×, 1 × MIC, and 8 mg/L (clinical breakpoint) of meropenem at 0, 4, 8, and 24 h. The green, red, and purple numbers show the average numbers of cells in the fields that have green, red, and total fluorescence, respectively.

**Figure 2 antibiotics-12-01170-f002:**
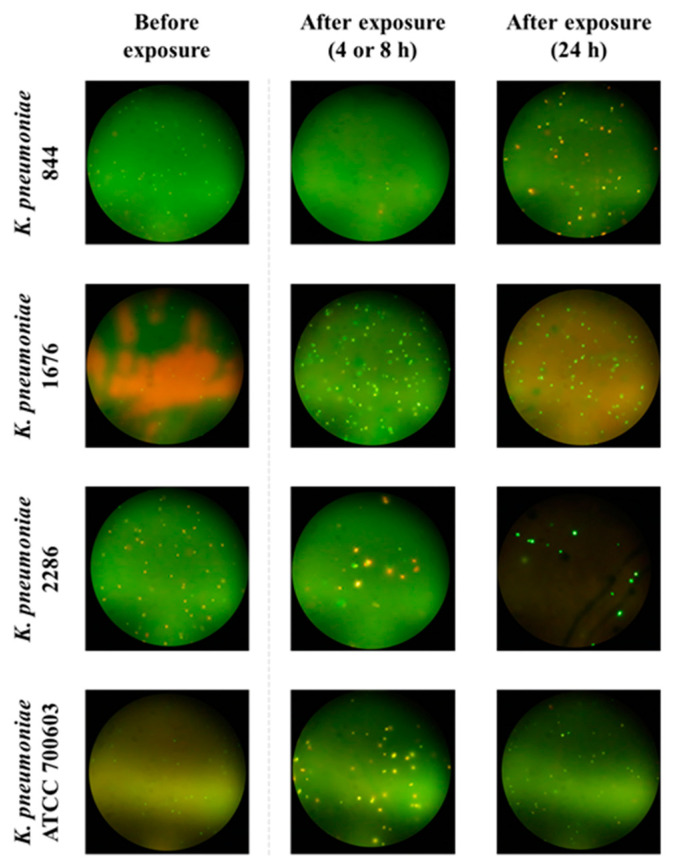
Typical microscope fields with altered cells of carbapenemase non-producing *K. pneumoniae* stains exposed to 0.5 × MIC of meropenem.

**Figure 3 antibiotics-12-01170-f003:**
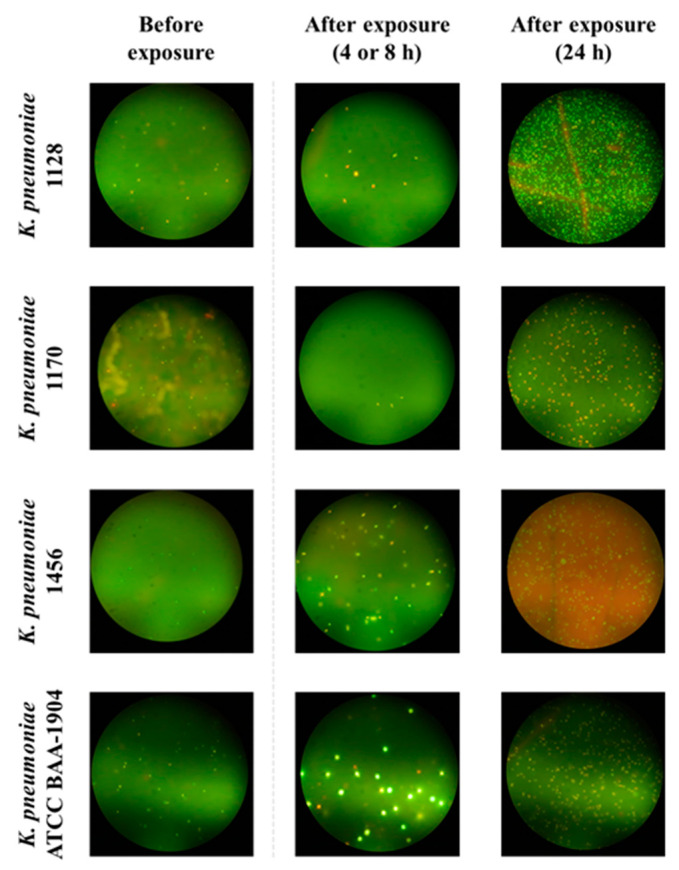
Typical microscope fields with altered cells of carbapenemase-producing *K. pneumoniae* stains exposed to 0.5 × MIC of meropenem.

**Figure 4 antibiotics-12-01170-f004:**
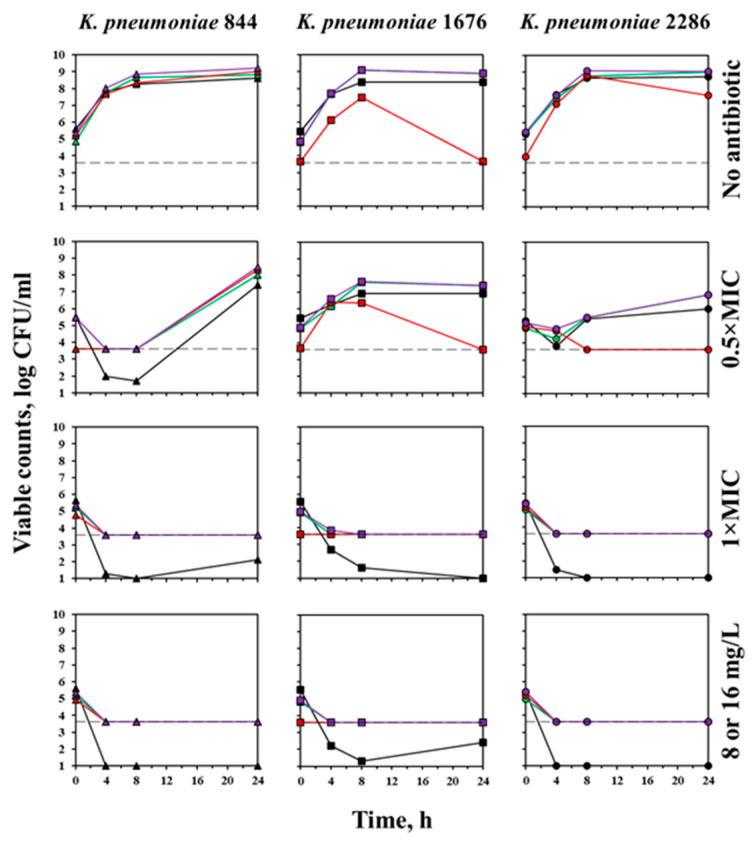
Time courses of carbapenemase non-producing *K. pneumoniae* in the absence of the antibiotic and exposed to 0.5×, 1 × MIC, and 8 mg/L (1676, 2286) or 16 mg/L (844) of meropenem. Black curves—results obtained from agar plating; green, red, and purple curves—indicate the results obtained from green, red, and total fluorescence, respectively. The dotted line indicates the limit of detection of microscopy (3.6 log (CFU/mL)).

**Figure 5 antibiotics-12-01170-f005:**
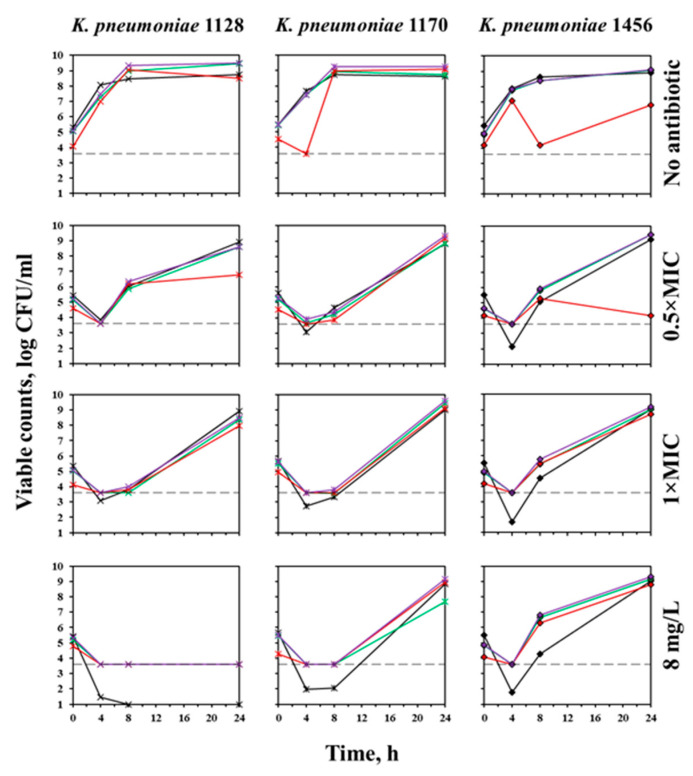
Time courses of OXA-48-producing *K. pneumoniae* in the absence of the antibiotic and exposed to 0.5×, 1 × MIC, and 8 mg/L of meropenem. Black curves—results obtained from agar plating; green, red, and purple curves—indicate the results obtained from green, red, and total fluorescence, respectively. The dotted line indicates the limit of detection of microscopy (3.6 log (CFU/mL)).

**Figure 6 antibiotics-12-01170-f006:**
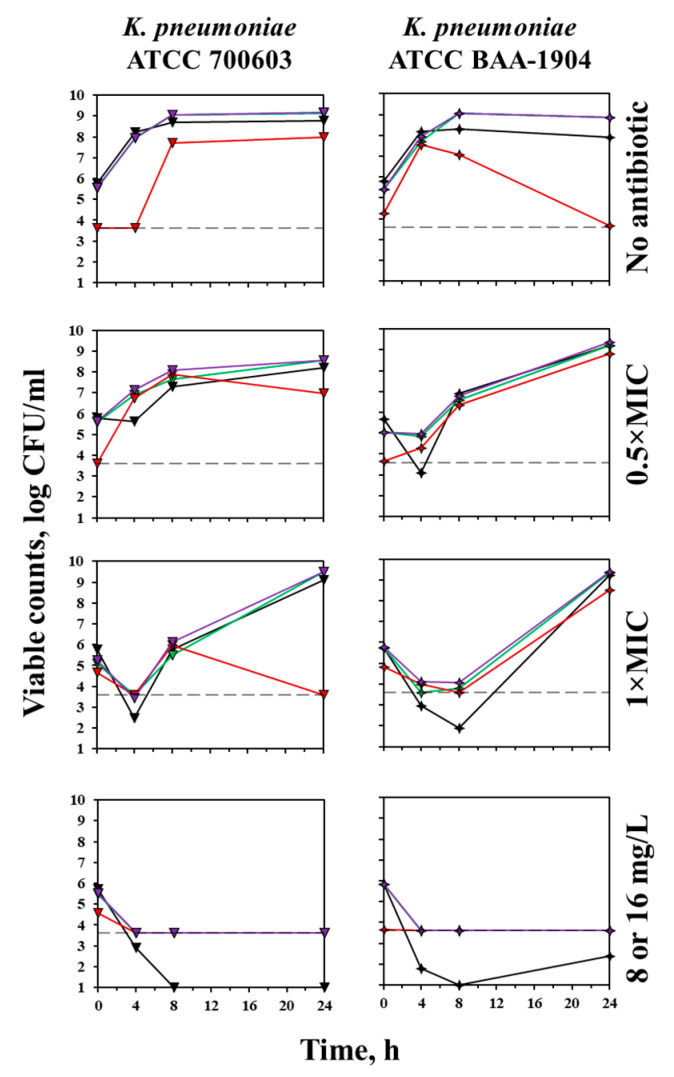
Time courses of carbapenemase non-producing *K. pneumoniae* ATCC 700603 and KPC-producing *K. pneumoniae* ATCC BAA-1904 in the absence of the antibiotic and exposed to 0.5 × MIC, 1 × MIC and 8 mg/L (former strain) or 16 mg/L (latter strain) of meropenem. Black curves—results obtained from agar plating; green, red, and purple curves—indicate the results obtained from green, red, and total fluorescence, respectively. The dotted line indicates the limit of detection of microscopy (3.6 log (CFU/mL)).

**Figure 7 antibiotics-12-01170-f007:**
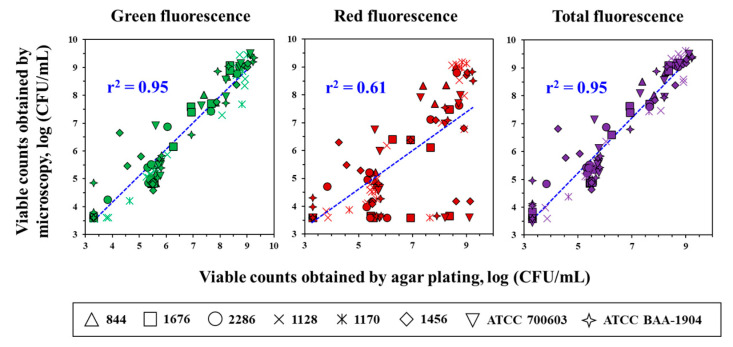
Correlation between viable counts of *K. pneumoniae* obtained via microscopy and agar plating at the same time points. Combined data on all eight *K. pneumoniae* strains fitted by Equation (1): *a* = 0.319, *b* = 0.961 (green fluorescence); *a* = 1.119, *b* = 0.696 (red fluorescence); *a* = 0.303, *b* = 0.985 (total fluorescence).

**Figure 8 antibiotics-12-01170-f008:**
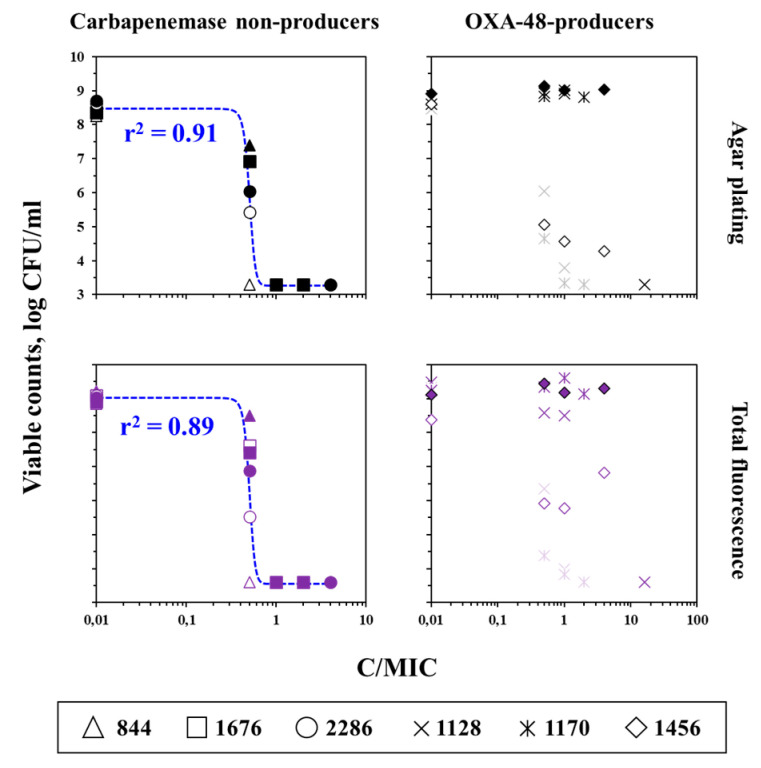
Correlation between viable counts of *K. pneumoniae* (combined data on three carbapenemase non-producing and three OXA-48-producing isolates) obtained at 8 h (empty or sheer symbols) and 24 h (filled symbols) and C/MIC. Data on carbapenemase non-producing isolates were fitted by Equation (2): *Y*_0_ = 3.3, *x*_0_ = 0.503, *a* = 5.188, *b* = −0.038 (upper left panel); *Y*_0_ = 3.6, *x*_0_ = 0.507, *a* = 5.431, *b* = −0.038 (lower left panel).

**Figure 9 antibiotics-12-01170-f009:**
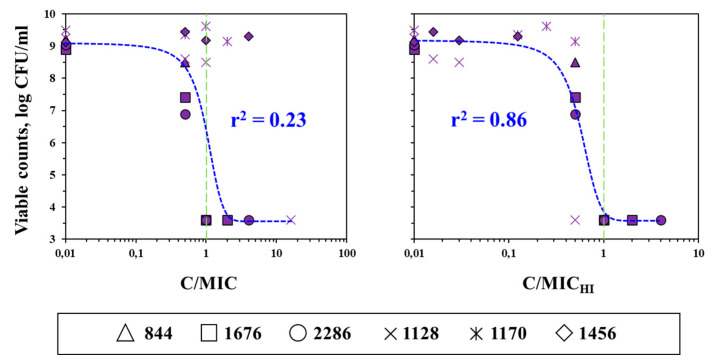
Correlation between viable counts of *K. pneumoniae* (combined data on three carbapenemase non-producing and three OXA-48-producing isolates, estimated by total fluorescence) obtained at 24 h and C/MIC or C/MIC_HI_. Data were fitted by Equation (2): *Y*_0_ = 3.6, *x*_0_ = 0.997, *a* = 5.72, *b* = −0.3 (left panel); *Y*_0_ = 3.6, *x*_0_ = 0.571, *a* = 5.733, *b* = −0.15 (right panel).

**Table 1 antibiotics-12-01170-t001:** Meropenem concentrations used for each *K. pneumoniae* strain.

*K. pneumoniae* Strain (Meropenem MIC, mg/L)	Meropenem Concentration, mg/L (Meropenem Concentration-to-MIC Ratio)			
	Flask 1	Flask 2	Flask 3	Flask 4
**844 (8)**	No antibiotic	4 (0.5)	8 (1)	16 (2)
**1676 (4)**		2 (0.5)	4 (1)	8 (2)
**2286 (2)**		1 (0.5)	2 (1)	8 (4)
**1128 (0.5)**		0.25 (0.5)	0.5 (1)	8 (16)
**1170 (4)**		2 (0.5)	4 (1)	8 (2)
**1456 (2)**		1 (0.5)	2 (1)	8 (4)
**ATCC 700603 (0.03)**		0.015 (0.5)	0.03 (1)	8 (267)
**ATCC BAA-1904 (8)**		4 (0.5)	8 (1)	16 (2)

## Data Availability

Not applicable.

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
