# Peer review of "Fluorescence Microscopy: Determination of Meropenem Activity against Klebsiella pneumoniae"

_antibiotics, 2023, doi:10.3390/antibiotics12071170_

Round 1
Reviewer 1 Report
The approach of this work is especially interesting since the methodology proposed to identify KP strains with very virulent potential due to their resistance phenotype is very useful to reduce the time currently used in the characterization techniques of these strains, which makes it possible to implement an effective early treatment, which could be very useful to reduce the mortality rate under these conditions of bacterial sepsis.
In this work there are some elements that must be considered before its publication.
In the text it says
1. DNA-based tests could provide only limited information on antibiotic susceptibility, since the knowledge of the resistance mutations and genes is limited and some mechanisms, which allow to withstand exposure to antibiotics, are non-genetic
Referring to this text of the document, it is interesting to remember that indeed many mechanisms associated with resistance to antibiotics are cryptic in nature. And therefore its molecular bases are not well described, so there is no treatment with specific antimicrobial drugs, which has the potential to increase health risks in the event of sepsis caused by particularly virulent strains, such as KP. I think it would be appropriate to include some reference to the resistance phenotype associated with cryptic genetic mechanisms.
This technique could be aimed at identifying quiescent KP cells responsible for reinfection processes, which are potentially dangerous since they generate outbreaks with appropriate resistance phenotypes in the presence of antibiotics.
2. In the text it says “Among the viable cells exposed to the antibiotic, there was a slight heterogeneity in the forms and intensity of fluorescence” the term slight heterogeneity should be more specific, or at least include in what aspect of cell morphology need to be detailed.
In the text later reference is made to size, in this sense if there are differences in size, it would be convenient to include a photograph where the difference could be measured or quantified. In this sense, it is worth considering that changes in cell size are associated with changes in molar concentrations under stress conditions, which modifies the physiological state of the bacterial cell. This approach, which reinforces the thesis that a morphological change is associated with a phenotypic change related to resistance, should be present in the discussion of the document.
3. In this work they used an antibiotic added to the culture plates (meropenem), and based on the added concentration they evaluated the process of (regrowth), in this sense I understand that it would have been appropriate to determine if the drug has a very heterogeneous distribution in the plate since its distribution could be altered by its diffusion coefficient, generating different concentration gradients at 8 and 24 hours, which could affect the regrowth process. Another hypothesis in this sense refers to the fact that the resistance mechanism or phenotype is present but must be adapted to the physical-chemical conditions of the environment, and during this period the energy flow is not directed towards reproduction but towards the maintenance and adaptation of expression regulation mechanisms.
In figure 2, it is not appreciated that they had included error bars. A measure of dispersion is always convenient, it would be convenient to explain the reasons why this methodology has not been used.
Figure 3 does not show the dispersion of the data with error bars either, I understand that it would be convenient to justify why this measure of data dispersion has not been used.
Regarding figure 4, it seems that the model fits the data reasonably well, however the equation does not appear in the document, it would be appropriate to include the mathematical model or adjustment equation so that future studies use the same adjustment system. This point is of special relevance because deviations from the fit with this model could tentatively serve to infer differences in the resistance phenotype and consequently infer potentially more virulent population increases.
Reviewer 2 Report
Dear Authors,
The manuscript ID: antibiotics-2443014 entitled: „Fluorescence Microscopy: Determination of Meropenem Activity Against Klebsiella Pneumoniae” is devoted to the rapid assessment of antibiotic activity.
In the recent years, antibiotic resistance is rising to dangerously high levels in all parts of the world. New resistance mechanisms (e.g. OXA-48-carbapenemases) are emerging and spreading globally threatening our ability to treat common infectious diseases. Therefore, the subject of this article is relevant and current. In this study, it was assessed whether fluorescence microscopy with acridine orange can be used to quickly assess (≤8 h) the activity of meropenem against Klebsiella pneumoniae. The whole manuscript (Introduction, Results, Discussion, Materials and Methods, and Conclusions) is properly organized. Introduction contains general data on the antimicrobial susceptibility testing. The obtained results are documented in the form of figures. Based on the results, discussion and conclusions were drawn.
However, I have some suggestions in order to improve paper, which are the following:
1) These studies were not performed on reference strains of K. pneumoniae. At the same time, the activity against the reference strains (one K. pneumoniae with OXA-48 and one K. pneumoniae strain without OXA-48) could be evaluated. The results would be more valuable and reliable.
2) Please add where the strains tested came from;
3) Other:
Lines 280, 288, 291: K. pneumoniae – with italics
Lines 285, 290: 5×105 CFU/mL – 5×105 CFU
Line 65: (MIC ≥ >8 mg/L)
Figure 1: 8 µg/ml and 8 mg/L – please unify the units
Line 200: (r2 0.95) – (r2 = 0.95)
Table 1: stain – strain
I agree with the Authors that the method proposed in the study needs improvement in terms of accelerating the growth and regrowth of bacteria for early meropenem MIC testing. According to me, this manuscript is interesting, but unfortunately these results are not sufficient to be accepted and published in such a prestigious journal as “Antibiotics”.
With highest regards,
Round 2
Reviewer 1 Report
Modifications made to the document cover comments and suggestions on my part